# Effects of Game Weekly Frequency on Subjective Training Load, Wellness, and Injury Rate in Male Elite Soccer Players

**DOI:** 10.3390/ijerph20010579

**Published:** 2022-12-29

**Authors:** Rim Sioud, Raouf Hammami, Javier Gene-Morales, Alvaro Juesas, Juan C. Colado, Roland van den Tillaar

**Affiliations:** 1Tunisian Research Laboratory “Sports Performance Optimization”, National Center of Medicine and Science in Sports (CNMSS), Higher Institute of Sport and Physical Education of Ksar Said, Manouba University, Tunis 2010, Tunisia; 2Research Laboratory, Education, Motor Skills, Sports and Health (LR19JS01), Higher Institute of Sport and Physical Education of Sfax, University of Sfax, Sfax 3029, Tunisia; 3Research Group Prevention and Health in Exercise and Sport (PHES), University of Valencia, 46010 Valencia, Spain; 4Department of Didactics of Musical, Plastic, and Body Expression, University of Valladolid, 47002 Valladolid, Spain; 5Department of Sport Sciences and Physical Education, Nord University, 7601 Levanger, Norway

**Keywords:** matches per week, rate of perceived exertion (RPE), profile of mood states (POMS), Hooper index

## Abstract

To compare the effects of playing one or two games per week on subjective perceived exertion (RPE) and (RPE-based) training load, monotony index, sleep, stress, fatigue, and muscle soreness (Hooper index), total mood disturbance, and injury rate in elite soccer players. Fourteen males from a first-division soccer club (age: 24.42 ± 4.80 years) competed in two games per week for six weeks and one game per week for twelve weeks (a total of 24 games). Paired *t*-tests and non-parametric Wilcoxon signed ranks evaluated the significance of the differences (*p* < 0.05). The main findings were that RPE was significantly larger when playing two games per week compared with one game. However, subject total and mean training load, mood disturbance, monotony, and subjective perception of sleep, stress, fatigue, muscle soreness monitoring (Hooper index), and the number of injuries were not different. The findings suggested that competing in two matches per week does not negatively influence injury rate and players’ perceptions of training load or wellness, even though players perceive two games per week as more physically demanding compared with one game per week.

## 1. Introduction

Soccer performance analysis is crucial and is usually related to a systematic change in the training load [1,2,3], in order to evaluate players’ achievement [4]. Training load variables, such as duration, frequency, and intensity, play an interactive role in improving and regulating physical capacity and performance [5]. In this context, researchers and coaches have defined soccer as a psychophysiological phenomenon [6,7], as the subtle changes in load and strain have a large effect on physiological and psychological variables in addition to wellness [2,6,8].

Over the last decades, researchers have utilized several psychological questionnaires aimed to monitor training stress, wellness, mood states, and strain to detect early signs of tiredness and/or overtraining and optimize high-level soccer training performance [2,9,10,11,12,13,14]. In this context, previous research [15,16,17] proposed a psychometric questionnaire based on self-assessment involving well-being ratings relative to sleep, stress, fatigue, and muscle soreness. Such a measure is useful in the longitudinal assessment of players’ perceptions over a season or longer. While previous studies have investigated the relationship between external and internal load and well-being in soccer players [18,19,20], the long-term effect of multiple matches played within a few days on related psychophysical parameters (i.e., perceived internal training load, feeling sensation, and injury rates) in professional soccer players is unknown.

Understanding how players’ perceptions of well-being, mood, and other parameters might change relative to the volume of soccer activity undertaken is key in managing players’ physical and psychological loads [21]. This is important because soccer players often must play up to two competitive matches in a week. In this sense, previous research [8,22] suggested that a recovery time of 72 to 96 h between two consecutive games is adequate to maintain physical performance. However, more is needed to reduce the injury rate compared to one game per week. Additionally, previous studies reported that players’ repeated jump and sprint performances are dropped immediately after a match [23,24], and ratings of muscle soreness remain elevated up to 72 h after 90 min of competitive soccer [25]. Similarly, Rollo, et al. [12] concluded that players’ ability to sprint, jump, and perform repeated intense exercise was impaired when playing two competitive matches a week over six weeks.

Given the importance of monitoring training load in soccer, the assessment of related variables has become commonly used in the literature and practice. Several studies [2,7,11] have supported the effectiveness of questionnaires for monitoring training-related perceived stress, wellness, strain, and recovery in players to detect early signs of tiredness and/or overtraining. Based on the available literature [26,27], it is reasonable to speculate that the physical performance of soccer players is reduced when they are involved in two or more weekly matches over prolonged periods. However, to our knowledge, no studies have examined how playing multiple soccer games in a week (i.e., one vs. two) may affect players’ well-being. Furthermore, more information on the effect of one vs. two playing games a week on subjective psychometric scores of training load and exertion (RPE), sleep, stress, fatigue, muscle soreness, mood state, and injury rate, is needed.

Therefore, this study aimed to analyze the differences between playing one or two competitive games a week on elite male footballers’ total and mean load, perceived exertion, monotony index, mood state disturbance, sleep, stress, fatigue, muscle soreness, and injury rate. Regarding the abovementioned, we hypothesized that playing two games per week compared to one would negatively affect the study’s dependent variables.

## 2. Materials and Methods

### 2.1. Experimental Approach to the Problem

This was a longitudinal design in which the effect of one versus two games per week was tested in elite male footballers’ total and mean load, perceived exertion, monotony index, mood state disturbance, sleep, stress, fatigue, muscle soreness, and injury rate. During each training session and competition, each player was given a questionnaire to gather data on mood state disturbance, sleep, stress, fatigue, and muscle soreness. The rate of perceived exertion (RPE) was collected 30 min after each game and training session [28]. The injury rate was monitored during the eighteen weeks of the study (24 games). Throughout the study, players were asked to maintain their routine nutritional habits.

### 2.2. Participants

The number of observations taken from each player (i.e., the number of games played in 1 game/week and that played in 2 games/week) may add biases to the study design. Therefore, an inclusion criterion of participating in at least 80% of the games of the study frame (with the same number of games in 1 game/week and 2 games/week) was established. Four participants were discarded from the study due to not complying with this criterion.

Finally, fourteen elite male soccer players (mean age: 24.42 ± 4.80 years, body mass: 78.62 ± 6.36 kg, height: 179.57 ± 7.01 cm; body mass index: 24.36 ± 1.11 kg/m^2^; body fat percentage: 13.04 ± 1.24%) belonging to one soccer club playing at the highest national division, participated in the study. All participants were experienced footballers (> than 7 years of competitive football experience) accustomed to physical training and regular competitive match play. All players were injury-free in the month before the start of the study. The players were fully informed of any possible risks and discomforts associated with the testing procedures and gave their written consent before participating. The present study was conducted according to the latest version of the Declaration of Helsinki. The protocol was fully approved by the Local Ethics Committee of the National Centre of Medicine and Science of Sports of Tunis (CNMSS-LR09SEP01).

### 2.3. Procedures

All the procedures were conducted in-season. During eighteen weeks divided into two periods, the 14 players participated in one official game (Sunday) or two (Sunday and Wednesday, around 72 h between games) per week. The first period comprised six weeks with two games per week and the second period, with one game per week, lasted twelve weeks to equate the total number of games. These two periods were separated by one month to avoid biases. They were engaged in 90 min training sessions of standard sport-specific training 5 times per week (passing, small-sided games, and ball shooting). During both designs, the training sessions comprised a warm-up, technical and tactical skill development as well as conditioning (aerobic and anaerobic training, speed, and agility training), and finally a cool-down.

All participants were familiarized with the training protocols prior to the investigation. After that, each Monday before the training session, all the players were provided with a questionnaire to gather data on mood state disturbance, sleep, stress, fatigue, and muscle soreness. The rate of perceived exertion (RPE) was collected 30 min after each game and training session [28]. The injury rate was monitored during the eighteen weeks of the study. The mean of all mood, load, injury rates, and RPE scores was calculated for each period and used for further analysis.

Each participant’s body height and mass were collected before starting the study using a wall-mounted stadiometer (Holtain Limited, Crosswell, Wales, UK) and an electronic scale (Electronics Development, New York, NY, USA), respectively. The sum of skinfolds was assessed using Harpenden skinfold calipers (Baty International, Burgess Hill, UK). Body measurements were conducted according to previous expert literature [29].

RPE was collected after each game and training session using the 10-point Borg’s scale [30] proposed by Foster, et al. [31]. This subjective estimation of the physiological load was obtained by asking each player: “How did you perceive your exertion during the training session?”. The use of the 10-point Borg’s scale modified by Foster, et al. [31] has largely been credited as the top five practical, cost-effective, and valid methods of quantifying internal training loads in soccer [32,33].

The session RPE-based training load was then calculated by multiplying the RPE value by the session duration [28,34]. The total weekly load obtained from 3 (two games a week) versus 5 (one game a week) training sessions was calculated as the sum of the training load of each training session of a given week. Then, the mean load calculated during the one-game-a-week condition for every 5 training sessions and during the two-game week period for 3 training sessions, was obtained by dividing the total load by the number of sessions of the week, and the standard deviation was also calculated [2,34]. 

The training monotony, measured every 2 days and calculated by the average over-time training variability, was calculated by dividing the mean load by the standard deviation [34]. 

Total mood disturbance was evaluated by the Profile of Mood States (POMS) questionnaire [35,36,37] after each training session and evaluated for the last 24 h. This self-report questionnaire consists of 65 adjectives aimed at assessing six states (anger-hostility, confusion-bewilderment, depression-dejection, fatigue-inertia, tension-anxiety, and vigor-activity) on a 5-point Likert scale, being 0 “not at all” and 4 “extremely”. Total mood disturbance was calculated as the sum of the five negative sub-scores minus the positive score plus a constant of 100 to avoid negative values. The following formula was used: Total mood disturbance = ([anger + confusion + depression + fatigue + tension] − vigor) + 100 Sleep, stress, fatigue, and muscle soreness monitoring

Upon arrival at the training session, players responded to questions extracted from Hooper questionnaires [17]. Questions included the players’ subjective perception about (i) the quality of sleep during the preceding night, (ii) quantity of stress, (iii) fatigue, and (iv) muscle soreness. Possible answers for each of these parameters ranged from “extremely low or good” (point 1) to “extremely high or bad” (point 7) [2]. The Hooper index was calculated as the addition of these four ratings.

The injury rate was calculated as all injuries obtained in the two conditions period (one vs. two games a week) and as an injury frequency during each mentioned period (training and matches) [8]. The effects of one vs. two matches per week on injury rate were analyzed for the 14 players during the study, which comprised eighteen weeks with 24 games. 

### 2.4. Statistical Analyses

All the statistical analyses were carried out using commercial software IBM SPSS Statistics for Macintosh (Version 27.0; IBM Corp., Armonk, NY, USA) and G*Power 3.0 [38]. The significance level for this study was established at *p* < 0.05. Results are reported as mean, standard deviation, and 95% confidence interval (CI).

The normality of data distribution was assessed using the Shapiro–Wilk test. Only the total mood disturbance and the hooper index showed a normal-gaussian distribution in both measurements (*p* > 0.05). It is also worth mentioning that the total load and mean load showed a normal-gaussian distribution only in the one-game per week measurement. All the rest of the variables (total load [in the two-day per week measurement], monotony index, mean load [in the two-day per week measurement], injury rate, and RPE) followed a non-normal distribution (*p* ≤ 0.05).

To assess differences between both game distributions (i.e., one game per week and two games per week) parametric paired *t*-tests and non-parametric Wilcoxon signed-ranks tests were used for the normally distributed and non-normal variables, respectively. The effect size was calculated as Cohen’s d with Hedges’ corrections to avoid biases due to sample size or standard deviation differences [39]. This corrected value is reported as unbiased Cohen’s d (dunb) [38], with dunb < 0.50 constituting a small effect, 0.50 ≤ dunb ≤ 0.79 a moderate effect, and dunb ≥ 0.80 a large effect [40]. 

Test–retest reliability of the variables was assessed using Cronbach’s model of intraclass correlation coefficient (ICCs) and standard error of measurements (SEM) according to the method of Hopkins [41]. According to Weir [42], we used ICC Model 3, k (i.e., the average of multiple scores of each subject measured by the same researchers).

## 3. Results

Table 1 displays the test–retest reliability analyses for all the tests conducted. Intraclass correlation coefficients (ICC) showed good reliability for all tests (values ranging from 0.86 to 0.91), with a standard error of measurement (SEM) from 1.37 to 3.49, and a coefficient of variation of <5%. Furthermore, paired *t*-tests showed no significant differences (*p* > 0.05) between the scores recorded during the two trials for all measured variables. 

Descriptive and inferential analyses of the dependent variables are presented in Table 2. The most notable finding was that the RPE was the only variable showing significant differences between the 2 games/week condition and 1 game/week condition (greater RPE when playing two games per week) with a large effect size (dunb = 11.04). On the other hand, no significant differences (*p* > 0.05) were observed in the rest of the study variables.

Figure 1 graphically presents the outcomes of the only variable (i.e., RPE) that presented significant differences between competing in one or two games per week.

## 4. Discussion

The present study examined the impact of playing one versus two competitive soccer games per week on the training load and perceived exertion, monotony index, mood state disturbance, hooper index, and injury rate in elite male soccer players. The main finding was that RPE was significantly larger when playing two games per week compared with one game. However, subjective load, total mood disturbance, monotony, subjective perception of sleep, stress, fatigue, injury rates, and muscle soreness monitoring (Hooper index) were not different between the two conditions. 

The elevated RPE scores in the two-game-a-week group compared to the one-game-a-week group were not unexpected. Prior work using RPE as a measure of subjective training load has reported this parameter to be sensitive to actual training loads undertaken and that higher RPE is observed when players undertake greater training loads [2,43,44]. The results of the current study in this respect are not new. However, they demonstrate, that competing twice per week is perceived as significantly more strenuous compared to one competition per week. 

Given the higher RPE reported for the two games per week condition, it was expected that there were also differences in subjective training load, which only revealed small effect sizes (see Table 2). This could be explained by the lower number of weekly training sessions per week (3 vs. 5) in the 2-games condition. This decreases the total and mean subjective training load. In addition, no significant effects were found in the total mood disturbance, monotony index, or hooper index (for sleep, stress, fatigue, and muscle soreness). This suggests that playing two games per week did not influence the player’s perception of training load much over the 2-games period, together with fatigue and muscle soreness. Thereby, this increase in the competition frequency did not compromise player wellness. Earlier similar studies with two competitions a week and three training sessions have also indicated that it does not affect performance variables after 72–96 h [8] nor stress and recovery indexes [12]. On the other hand, another study reported that sprint, jump, and intense exercise performance were impaired 48 h post-match when players engaged in around two competitive games per week over six weeks [12,21]. Additionally, previous expert literature reported that periods with congested match playing can decrease physical and mental fitness in elite soccer players, relying these changes on psychological rather than hormonal parameters [21]. 

Regarding the injury rate, our results demonstrated no significant effects of playing twice-a-week games compared to once-a-week, which could be a key observation of the present study. Prior studies examining the effects of different frequencies of competitive games on physical performance variables have reported significantly higher muscle damage [42,45] and injury rates [8] with increasing games per week. In this sense, it is worth highlighting that different approaches to what an injury is may result in varying results within the literature [46]. Rollo, et al. [12] noted that a lack of data regarding injury risk hampers conclusions about game frequency and player performance. In this regard, our results regarding the invariable injury rate with two games a week only refer to six weeks. Therefore, two games per week for longer than six weeks may increase cumulative injury risk. Research examining this issue is necessary to draw robust conclusions. 

There are more limitations in the study besides the short period of six weeks. Although the players maintained the same volume of training outside games across the whole study, the lack of monitoring other aspects of physical load (e.g., heart rate, lactate) experienced by the participants during the study is a limitation. Likewise, we did not assess dietary intake during the period and nutritional practices can influence performance and perception of performance and related factors in soccer [47]. Therefore, future work examining training load and/or player wellness over a prolonged period should also consider incorporating physiological load parameters and dietary assessment. Finally, in the present study, the congested fixture took place in the first part of the study. A congested fixture is supposed to affect players’ performance only if they play all games. To control this and reduce potential biases, we established an inclusion criterion of participating in at least 80% of the games of the study (same number of games in 1 game/week and 2 games/week). According to the procedures of the present study, other statistical procedures such as generalized estimating equation (GEE) could provide further details on players’ responses to the weekly frequency of games. 

Despite the mentioned limitations and while caution should be applied until more scientific evidence arrives, the present study strengthens the idea that elite soccer athletes can compete in two games per week without having higher injury rates or subjective training load, or lower wellness. Playing two games per week negatively affected the physical demands perception of the athletes.

## 5. Conclusions

Playing two competitive matches per week over six weeks does not significantly influence subjective training load, wellness, and injury incidence. On the other hand, players do perceive two games a week as more physically demanding (via RPE). 

## Figures and Tables

**Figure 1 ijerph-20-00579-f001:**
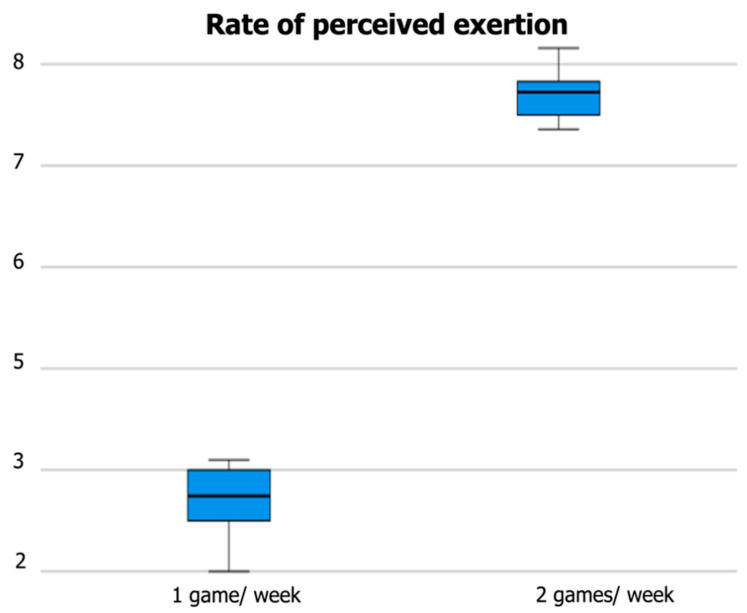
Rate of perceived exertion (RPE) values.

**Table 1 ijerph-20-00579-t001:** Test–retest reliability of the variables.

	ICC (95% CI)	SEM	CV (%)
Rate of perceived exertion	0.90 (0.87–0.94)	1.37	1.85
Total load	0.87 (0.79–0.89)	2.37	2.15
Mean load	0.90 (0.83–0.92)	3.28	1.53
Monotony index	0.88 (0.82–0.90)	2.78	2.16
Total mood disturbance	0.91 (0.82–0.93)	3.49	2.55
Hooper index	0.86 (0.78–0.89)	2.47	3.17

ICC: intraclass correlation coefficient; CI: confidence interval; SEM: standard error of measurement; CV: coefficient of variation.

**Table 2 ijerph-20-00579-t002:** Mean, standard deviation (Std. Deviation), and 95% confidence interval (CI) of the dependent variables. Additionally, the significance of the differences between participating in one and two games per week in elite soccer players.

Variables		Mean	Std. Deviation	95% CI		*p*-Value	Cohen’s Dunb
				Lower	Upper		
RPE	1 game/week	3.62	0.39	3.40	3.85	<0.001 *	11.04 (large)
	2 games/week	7.73	0.24	7.59	7.88		
Total load	1 game/week	1543.97	287.51	1377.97	1709.98	0.096	0.48 (small)
	2 games/week	1829.91	529.83	1523.99	2135.83		
Mean load	1 game/week	578.71	53.33	547.92	609.51	0.064	0.41 (small)
	2 games/week	614.57	72.74	572.57	656.57		
Monotony index	1 game/week	2.50	0.76	2.06	2.94	0.587	0.15 (small)
	2 games/week	2.71	0.91	2.19	3.24		
Total mood disturbance	1 game/week	45.42	13.25	37.77	53.06	0.101	0.36 (small)
	2 games/week	48.88	13.48	41.10	56.66		
Hooper index	1 game/week	10.48	1.70	9.52	11.46	0.137	0.31 (small)
	2 games/week	11.00	1.55	10.10	11.90		
Injury rate	1 game/week	0.57	0.65	0.20	0.94	0.059	0.56 (moderate)
	2 games/week	0.21	0.43	0.03	0.46		

* Indicates a significant level on *p* < 0.05.

## Data Availability

The data presented in this study are available on reasonable request from the corresponding author.

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
