# Peer review of "Effects of Game Weekly Frequency on Subjective Training Load, Wellness, and Injury Rate in Male Elite Soccer Players"

_ijerph, 2022, doi:10.3390/ijerph20010579_

Round 1

Reviewer 1 Report

Effects of game weekly frequency on subjective training load, wellness, and injury rate in male elite soccer players

First of all, the reviewer would like to thank the authors for their work and efforts in trying to improve sports science knowledge.

General comments to the authors

The article is investigating the effects of playing one or two games per week on subjective perceived exertion (RPE) and (RPE-based) training load, monotony index, sleep, stress, fatigue, and muscle soreness (Hooper index), total mood disturbance, and injury rate in elite soccer players. Overall, the study is well designed and well-written, with a great introduction proposing the usefulness of the topic and a clear outline of the research question. I suggest that the author modify/include some suggestions in order to improve the manuscript prior to be published:

Abstract

This section is well designed and well-written.

Introduction section

I think that there are a lot of out outdated articles used in introduction section. Because of the fact that the authors should use this recent article to support their ideas for soccer players’ training loads and wellness in introduction and discussion sections.

Oliva Lozano, J. M., Muyor, J. M., Pérez-Guerra, A., Gómez, D., Garcia-Unanue, J., Sanchez-Sanchez, J., & Felipe, J. L. (2022). Effect of the Length of the Microcycle on the Daily External Load, Fatigue, Sleep Quality, Stress, and Muscle Soreness of Professional Soccer Players: A Full-Season Study. Sports Health, 19417381221131531.

Nobari, H., Arslan, E., Martins, A. D., & Oliveira, R. (2022). Are acute: chronic workload ratios of perceived exertion and running based variables sensible to detect variations between player positions over the season? A soccer team study. BMC Sports Science, Medicine and Rehabilitation, 14(1), 1-14.

Maupin, D., Schram, B., Canetti, E., & Orr, R. (2020). The relationship between acute: chronic workload ratios and injury risk in sports: a systematic review. Open access journal of sports medicine, 11, 51.

Clemente, F., Silva, R., Arslan, E., Aquino, R., Castillo, D., & Mendes, B. (2021). The effects of congested fixture periods on distance-based workload indices: A full-season study in professional soccer players. Biology of Sport, 38(1), 37-44.

Methods section

L-113: Please give information about their usual training regime as contents and duration.

L-113: Please give information about their usual warm-up as contents and duration.

Statistical Analyses

Cohen’s d with its descriptors, such as trivial, small and moderate, should be added Statistical Analyses, Table 2 and 3.

Discussion section

Overall the discussion is well-written and incorporates relevant literature. The authors should add sample size as a limitation. What about their strength side of the study. Is there any. Please give some information about it.

Tables and Figures

These sections are well designed and well-written. However, the authors can make a figure regarding rpe and total mood disturbance responses to compare 1 game in a week and 2 games in a week to understand more easily for readers.

Author Response

See file

Reviewer 2 Report

Introduction

P2L48-50: It is important to be more specific on the gap as previous studies have addressed very similar topics to what is pointed out in this sentence:

https://www.mdpi.com/2079-7737/11/3/467

https://www.frontiersin.org/articles/10.3389/fpsyg.2021.671072/full

https://www.sciencedirect.com/science/article/pii/S0031938420304315

The introduction does not provide full information on the effect of congested fixtures on the players in a soccer context. However, the vast literature on this topic allows a deeper discussion to support the hypothesis. For example, P2L67-69: what are the cited studies that support this assumption?

The dependent variables are not introduced in the introduction. It is not clear why these specific variables were selected instead of others (and why all of them are required).

L91: it is not clear how the sample size estimation used both literature data and data from a pilot study. When computing the sample size, just one value is added (or the authors merged the values?). Also, it is not clear what variable was used and which settings were adopted for the sample size estimation. Finally, if it was done in GPower, please mention the software.

L115: please indicate how the players got familiar with the questionnaires.

L168: it is unclear what the authors call “tendencies”. If close to 0.05 p-values are assumed as significant differences in the current study, please state it clearly as there seems to be no literature supporting classification of p-values.

Regarding the experimental design, it is not clear how the congested fixture was analyzed. A congested fixture is supposed to affect players' performance only if they play both games. As you have 18 players, it is expected that some of them did not play all the matches. Therefore, it is not clear how you decided whether to include or not a player within the sample.

Also, as paired t-tests were adopted, it is important to indicate how these data pairs were created. Even if the player had played all the matches, he engaged in 24 games. Therefore, there are 12 pairs of data. How were they grouped? The arbitrary choice of pairing the first congested and non-congested fixture is insufficient to explain the analysis (as they were distant in time). Finally, if there are multiple measures from a single player, t-tests are not recommended, as there is no control. Maybe a different statistical analysis is required (maybe a GEE)

Results

This section is started with reliability measures. However, there are no details in the methods section about it. I suggest including it.

Discussion

The information presented on the injury rate (L212) differs from the information presented in the results section (L199)

L238: it is not clear what result of the present study supports this affirmation.

L240: a non-significant impact is the same as the absence of impact? Again, there seems to be confusion when interpreting p-values or trying to arbitrarily classify them (searching for “tendencies”).

Reviewer 3 Report

The research presented in this article was scientifically well conceptualized and executed. However, some areas require attention to bring them to a level of publication. The authors should attend to the following to edit into papers.

Author Response

See file

Round 2

Reviewer 2 Report

Dear authors,

Indeed, the writing is much better, and the article has improved greatly.

However, the data analysis still requires explanation. You have mentioned a paired t-test as the statistical analysis. However, pairing samples requires them to have the same conditions. For example, if player A engaged in 5 non-congested and 6 congested matches, how did you pair the data (when organizing them in the SPSS spreadsheet, data in the same line is paired). Also, multiple observations from single subjects bias the analysis, which requires a GEE instead of a t-test. To avoid this bias, all the subjects should have the exact number of observations (what we know is impossible in the real world of soccer). That's why t-tests are not recommended.

As you have some p-values really close to the critical point (0.042, 0.059), conclusions might change depending on the data analysis.

Finally, please indicate which ICC was used (see Weir, 2005 - https://pubmed.ncbi.nlm.nih.gov/15705040/).

Author Response

Point-by-point statements to the reviewers’ comments

We thank our reviewer for the time spent re-reviewing the article and
suggesting ways to improve the robustness of the analyses of our manuscript We have been reading about how to conduct GEE. We have tried to conduct the analysis, but we believe we do not have enough data for that (e.g., between-subject variables).

Additionally, although we are not experts in this kind of analysis, we found that GEE is generally used to predict variables more than to compare conditions.

To control for the uneven number of observations we have selected the players who participated in at least 80% of the games (with the same number of games in 1 game/week and 2 games/week). We have included a sentence in the limitations section indicating that GEE could provide more specific data for this study design.

With the new analyses, as the reviewer pointed out, the results have varied and the significance close to critical point have changed. We sincerely appreciate this suggestion.

We have also specified that we used Model 3,k ICC according to Weir (2005).